# The Waning of BNT162b2 Vaccine Effectiveness for SARS-CoV-2 Infection Prevention over Time: A Test-Negative Study in Health Care Professionals of a Health Department from January 2021 to December 2021

**DOI:** 10.3390/ijerph192113884

**Published:** 2022-10-25

**Authors:** Natali Jiménez-Sepúlveda, Pablo Chico-Sánchez, José Miguel Castro-García, Isabel Escribano-Cañadas, Esperanza Merino-Lucas, Elena Ronda-Pérez, José Sánchez-Payá, Paula Gras-Valentí

**Affiliations:** 1Epidemiology Unit, Department of Preventive Medicine, Alicante Institute for Health and Biomedical Research (ISABIAL), General University Hospital Dr. Balmis, 03010 Alicante, Spain; 2Department of Medical Imaging, General University Hospital Dr. Balmis, 03010 Alicante, Spain; 3Department of Microbiology, Alicante Institute for Health and Biomedical Research (ISABIAL), General University Hospital Dr. Balmis, 03010 Alicante, Spain; 4Unit of Infectious Diseases, Alicante Institute for Health and Biomedical Research (ISABIAL), General University Hospital Dr. Balmis, 03010 Alicante, Spain; 5Preventive Medicine and Public Health Area, Faculty of Health Sciences, University of Alicante, 03690 Alicante, Spain; 6Centre of Networked Biomedical Research in Epidemiology and Public Health (CIBERESP), 28029 Madrid, Spain

**Keywords:** COVID-19, SARS-CoV-2, SARS-CoV-2 vaccine, negative cases and controls, vaccine effectiveness, healthcare personnel, infection prevention and control

## Abstract

The duration of protection of vaccines against SARS-CoV-2 infection has been evaluated in previous studies, but uncertainty remains about the persistence of effectiveness over time and the ideal timing for booster doses. Therefore, the aim of this study was to evaluate BNT162b2 vaccine effectiveness against SARS-CoV-2 infection in health care workers (HCWs) at a tertiary hospital depending on time elapsed since the completion of a two-dose vaccination regimen. We conducted a case–control with negative test study between 25 January and 12 December 2021 that included 1404 HCWs who underwent an active infection diagnostic test (AIDT) to rule out SARS-CoV-2 infection due to COVID-19 suspicion or prior close contact with patients diagnosed with COVID-19. The adjusted vaccine effectiveness (aVE) for the prevention of SARS-CoV-2 infection 12 to 120 days after completing the full two-dose vaccination regimen was 91.9%. Then, aVE decreased to 63.7% between 121 to 240 days after completing the full two-dose regimen and to 37.2% after 241 days since the second dose. Vaccination against SARS-CoV-2 infection in HCWs remains highly effective after 12 to 120 days have elapsed since the administration of two doses of the BNT162b2 vaccine; however, effectiveness decreases as time elapses since its administration.

## 1. Introduction

The World Health Organization (WHO) declared, in March 2020, the novel coronavirus disease (COVID-19) a global pandemic affecting more than five hundred seventy million and causing more than six million deaths [1,2]. The initial nonpharmacological measures for outbreak control were not enough, highlighting the importance of initiating early vaccination strategies aimed at reducing illness severity and mortality. These strategies led to a decrease in the health care burden caused by the disease and have become a cornerstone for disease containment [3,4].

The vaccination campaign against SARS-CoV-2 infection in Spain began on 27 December 2020 aimed at nursing home residents and staff, as well as front-line health care workers (HCWs) [5]. It was later introduced to the general population, prioritizing the elderly.

By 30 July 2022, in Spain more than 100 million doses had been administered, representing a coverage of 85.8% with a complete two-dose vaccination regimen [6]. Globally, more than twelve billion doses have been administered, representing a population coverage of 62.4% [7]. The close monitoring of mRNA vaccines against COVID-19 has permitted evaluating their effectiveness in a real-word setting, proving vaccine effectiveness (VE) of over 90% [8]. In this regard, the European Medicines Agency (EMA) and the European Centre for Disease Prevention and Control (ECDC) have reinforced the importance of expanding our knowledge of VE beyond clinical trials. These types of studies are essential for generating evidence for the continuous evaluation of vaccine benefits and risks and improved decision making, including vaccination strategies aimed at general populations or at priority groups, such as HCWs.

Some previous studies have provided further evidence of the initial VE results by evaluating it in different populations and analyzing different clinical outcomes [9,10,11,12].

Studies conducted in HCWs within the Infection Prevention and Control Program of a health department (HD) obtained VE for SARS-CoV-2 infection prevention after the first and second doses of 96.3% (95% CI: 82.5–99.2%) and 52.6% (95% CI: 1.1–77.3), respectively [13,14]. Similarly, another study evaluating VE during a long-term follow-up in participants within a randomized phase 2–3 trial of BNT162b2 vaccine showed a reduction in VE from 96% (for a period from 7 days to 2 months after receiving the second dose) to 84% (for a period of 4 months to approximately 7 months after receiving the second dose) [15]. These findings reinforce the need for further research studies that evaluate VE both for infection prevention and for its performance depending on time elapsed since its administration.

Therefore, our purpose was to evaluate the effectiveness of BNT162b2 vaccine for SARS-CoV-2 infection prevention in a group of HCWs in a HD after finishing a complete two-dose regimen, depending on time elapsed since its administration.

## 2. Materials and Methods

A case–control study with negative test was carried out with health care workers (HCWs) of a health department (consisting of twelve primary care centers and a tertiary hospital, attending to over 285,000 occupants, that is also a referral hospital for surrounding communities with a total population of about 2 million people) between the epidemiological weeks 4 to 29 (from 25 January to 12 December 2021).

The vaccination campaign against SARS-CoV-2 with BNT162b2 and mRNA-1273 vaccines aimed at HCWs was initiated on 8 January 2021, under a protocol in compliance with the technical specifications outlined by the vaccine datasheet [16]. We included those HCWs who were tested with an AIDT (either a PCR or an antigen test) for having symptoms consistent with COVID-19 or being close contacts of a confirmed case. HCWs with a positive AIDT result were considered cases, while those with negative results were considered controls.

Data collection was carried out by the preventive medicine department through telephone interview and through specific information retrieval from all medical records, such as age, sex, hospital department affiliation, professional category, date of symptom onset, date of AIDT and previous history of PCR confirmed COVID-19.

Vaccination status was obtained from the Nominal Vaccine Register (NVR). The NVR is a database containing all the records of the vaccines administered to each citizen receiving health assistance in the Valencian community (Spain). Symptomatic HCWs who had received both doses and whose symptoms started at least 7 days after vaccination were considered vaccinated with a complete regimen. HCWs who had received only the first dose and whose symptoms started at least 12 days after vaccination were considered incompletely vaccinated. Those who had received both doses and less than 12 days had elapsed since symptoms onset were also considered incompletely vaccinated. Asymptomatic HCWs who had contact with confirmed cases were considered vaccinated with a complete regimen if they had received the second dose at least 7 days before the PCR test was performed. They were considered vaccinated with an incomplete regimen if they had received only the first dose and at least 12 days had elapsed since the PCR test was performed. HCWs who had previously contracted COVID-19 were considered vaccinated with a complete regimen if they had received one dose and at least 7 days had elapsed since the onset of symptoms or since the PCR test was performed. Finally, those who did not meet any of the above criteria were considered unvaccinated. For time-dependent VE assessment, three periods were established depending on time elapsed between the administration of the second dose and AIDT completion: the first one comprised the period between 12 to 120 days, the second one from 121 to 240 days and the last one the period after 240 days.

The study was approved by the Drug Research Ethics Committee of the Department of Health of the General University Hospital of Alicante with code PI2021-088.

### Statistical Analysis

Health care workers’ (HCWs) characteristics were described according to their SARS-CoV-2 vaccination status (complete or incomplete) and according to the time elapsed between vaccination and AIDT completion (12 to 120 days, 121 to 240 days and more than 241 days). The different groups were compared using the chi-squared test. The associations between SARS-CoV-2 infection, vaccination status and other variables (age, sex, professional category, professional setting, and different comorbidities) were analyzed by calculating the unadjusted odds ratio (OR) and adjusted odds ratio (aOR) through a logistic regression model. Variables that showed statistically significant differences among the three groups (according to time elapsed since vaccination), as well as those significantly associated with SARS-CoV-2 infection, were included in the logistic regression model. The adjusted vaccine effectiveness (aVE) was calculated using the formula aVE = (1 − aOR) × 100 with a 95% confidence interval (95% CI), both for the overall HCWs and for different subgroups depending on age, sex and cause of enrollment (having symptoms consistent with COVID-19 or being close contacts of a confirmed case). Statistical significance was set to *p* < 0.05 and the analysis was performed using SPSS software (version 25.0; IBM Corp., Armonk, NY, USA).

## 3. Results

A total of 1404 health care workers (HCWs) were included in this study; of these, 139 (9.9%) were not vaccinated, 284 (20.2%) were incompletely vaccinated and 981 (69.9%) were vaccinated with a complete regimen. Taking into account the time elapsed between vaccination and AIDT completion, 223 (15.9%) HCWs were vaccinated between 12 and 120 days, 466 (33.2%) between 121 and 240 days and 292 (20.8%) after more than 241 days (Table 1).

Of the 1440 HCWs included in the study, 209 (14.9%) were considered COVID-19 cases and 1195 (85.1%) were considered controls. The frequency of exposure to the complete vaccination schedule between 12 and 120 days was 2.9% in COVID-19 cases versus 18.2% in controls (*p* < 0.001). For HCWs who received the complete vaccination regimen when more than 241 days had elapsed, the aVE was 25.5% in COVID-19 cases versus 20.0% in controls (*p* = 0.092). See Table 2.

Adjusted vaccine effectiveness (aVE) after 12 to 120 days was 91.9% (95% CI: 79.8–96.8); between 121 to 240 days, 63.7% (95% CI: 40.4–77.9); and after 241 days, 37.2% (95% CI: −5.1–62.5). The majority of participants were women (1064 (75.8%)), showing a decrease in aVE after 12 to 120 days since the second dose administration from 89.8% (73.6–96.1%) to 45.4% (1.0–69.9%) after more than 241 days had elapsed. Likewise, those older than 40 years (783 (55.8%)) showed a decrease in aVE from 90.1% (68.4–96.9%) at 12 to 120 days after administration of the complete regimen to 39.2% (−23.8–70.1%) after 241 days. Regarding the motive of study, 909 (64.7%) HCWs were under suspected disease and showed a decrease in aVE from 92.1% (64.3–98.2%) to 2.0% (−100.6–52.1%) for the periods between 12 to 120 days and after 240 days, respectively. See Table 3.

## 4. Discussion

Our results demonstrate that vaccine effectiveness (VE) against SARS-CoV-2 infection in HCWs decreased over time since the completion of a two-dose vaccination regimen with BNT162b2 from 91.9% between 12 to 120 days after vaccination to 37.2% after 240 days.

Since vaccination against SARS-CoV-2 infection was initiated, multiple studies have evaluated VE over time [17,18,19]. Our results are consistent with previous studies such as the one performed by Bedston et al., in which a decrease in VE ranging from 86% to 53% was found for periods between 14 to 154 days after completing a two-dose vaccination regimen [20]. Another study carried out in Finland showed that VE after 14 days after receiving a second dose was 80%, decreasing to 53% after 90 days [21]. Although these results are not identical to ours, this could be explained by the different follow-up periods evaluated by each: In the first study, effectiveness over time was evaluated over five months and in the second over three months, while in ours, the follow-up period comprised more than eight months. Despite having demonstrated that vaccination against SARS-CoV-2 infection is highly effective against COVID-19, other causes related to the vaccines’ waning effectiveness have been studied. The WHO, in conjunction with partners, expert networks, national authorities, health care institutions and researchers, has been monitoring and evaluating the SARS-CoV-2 outbreak since January 2020. The emerging variants that showed an increased risk for global public health during late 2020 prompted the introduction of the specific categories variant of interest (VOI) and variant of concern (VOC), in order to prioritize monitoring and research on a global scale and to guide specific actions against the COVID-19 pandemic [22]. These variants are associated with an increased risk of transmission and severity, a reduction in microbiological diagnostic test sensitivity and worse response to treatment due to weak vaccine-mediated immune response [23].

Our study was conducted between epidemiological weeks (EW) 4 and 49 (from January to December 2021). During the first half of this period, the alpha variant prevailed in Spain, whereas after EW 27, the delta variant became the dominant one, causing more than 50% of the PCR-confirmed COVID-19 cases and reaching 90% predominance from EW 29 to 49 [24].

During EW 33 (18 August 2021), the CDC (Centers for Disease Control and Prevention) published a study regarding a BNT162b2 and mRNA-1273 vaccine effectiveness assessment for SARS-CoV-2 infection among nursing home residents before and during the widespread transmission of the delta variant between 1 March and 1 August 2021.

The results showed a decline in VE ranging from 74.7% at the beginning of the study (March–May 2021) to 53.1% during the period wherein the delta variant had the highest prevalence (June to July 2021) [25]. This study differs slightly from our results; however, it should be considered that the period evaluated in this study was shorter than ours (5 months vs. 8 months) and that its sample included frail elderly. We, on the other hand, included HCWs aged between 18 and 65 years. There is currently considerable scientific evidence regarding VE reduction in the elderly population, in whom a higher risk of morbidity, hospital admissions and death due to COVID-19 have been reported, further increasing among those who are institutionalized [26,27]. Another observational study published by Fabiani et al. in Italy evaluated VE in a group of patients over 16 years old during a similar period to our study (from December 2020 to November 2021) in which the predominant variants were alpha and delta. Their results indicate a decrease in VE from 82% to 33% between 28 and 210 days after the completion of a two-dose vaccination regimen [28]. These results are very similar to those found in our study, where the delta variant prevailed between EW 26 and 29 and during which we found a drop in VE from 91.9% to 37.2% from 12 to more than 241 days after the second dose [28].

In Spain, a booster dose was included in the vaccination strategy starting on 27 December 2021, prioritizing patients in nursing homes, those over 60 years old, health care and social care workers and those who received a first dose with adenovirus vaccines (Janssen or Vaxzevria) [29]. Although our study evaluated vaccine effectiveness after two doses depending on time elapsed since vaccination, it also aimed to emphasize the importance of and need for administering additional booster doses, as appropriate and in compliance with SARS-CoV-2 infection prevention and control strategies. Hence, given the evidence of waning vaccine effectiveness due to multiple factors, efforts to provide maximum vaccination coverage by administering additional booster doses, when appropriate, are critical. The findings of this study are subject to certain limitations. Firstly, there may be a selection bias due to the single inclusion of the BNT162b2 vaccine, without considering other vaccines authorized by the time the study was performed. However, the BNT162b2 vaccine was the first one used in HCWs during early vaccination campaigns and the one that most of them received. Secondly, the specific variant that infected each of the HCWs included could not be determined in all cases, so we were unable to evaluate stratified VE in this regard. This was because variant analysis was only performed in some randomly selected cases. Nevertheless, as discussed above, during the period analyzed, the delta variant prevailed, representing more than 50% of the diagnosed cases. Additionally, an unpublished study under editorial evaluation [30] conducted in HCWs from the European region of the Valencian community between EW 4 and 28 showed a drop in VE during the periods from 12 to 120 days and more than 120 days after the second dose administration from 91.6% to 71.5%. During the period analyzed in this study, the alpha variant was the prevailing one, a fact that highlights that VE decline occurred in a period that preceded the predominance of the delta variant. These findings suggest that the decrease in VE depends on time elapsed since vaccination regardless of the emergence of new SARS-CoV-2 variants. Thirdly, there was no representative sample of subjects over 65 or under 18 years old, although it should be noted that since active HCWs’ age is almost completely included within this range, the entire population at risk was analyzed in our study. Finally, we did not include in the analysis of the aVE (Table 3) the following variables: professional category, hospital department and previous COVID-19 infection because we did not have enough data for their statistical analysis, which could be a limitation. However, these variables do not biologically predispose to SARS-CoV-2 infection in health care workers.

The strengths of this study are, on one hand, the use of a case–control study design with negative test, which is the standard recommended by the WHO for the analysis of VE against both SARS-CoV-2 and seasonal influenza infections [31,32]. This design minimizes selection bias related to seeking medical care. On the other hand, the ease of access to the preventive medicine consultation by HCWs, implemented since the beginning of the pandemic, has allowed for the rapid care and diagnosis of both severe and mild cases of SARS-CoV-2 infection. This might have reduced the selection bias by counteracting the reduced demand for medical assistance due to mild or nonspecific symptoms associated with SARS-CoV-2 infection. In addition, the Epidemiological Surveillance System implemented by the preventive medicine department of our hospital, in concert with health authorities, has allowed a continuous and rigorous evaluation of HCWs in our health department, ensuring that HCW casualties are practically negligible and preventing VE from being affected.

## 5. Conclusions

Our findings show that VE decreases progressively 4 months after the administration of the complete regimen with two doses of BNT162b2 vaccine against SARS-CoV-2 infection in HCWs. This not only reinforces the importance of administering a booster dose and prioritizing groups at risk but also highlights the need for continuous VE monitoring in HCWs in order to assess the endurance of the protection provided by the current strategies.

## Figures and Tables

**Table 1 ijerph-19-13884-t001:** Characteristics of the health care workers included in this study according to the vaccination regimen they received (n = 1404).

	Total n (%)	Complete >241 Days n (%)	Complete 121–240 Days n (%)	Complete 12–120 Days n (%)	Incomplete n (%)	Unvaccinated n (%)	*p*-Value ^d^
Total	1404 (100)	292 (20.8)	466 (33.2)	223 (15.9)	284 (20.2)	139 (9.9)	
**Sex**							0.485
Male	340 (24.2)	78 (26.7)	105 (22.5)	60 (26.9)	68 (23.9)	29 (20.9)	
Female	1064 (75.8)	214 (73.3)	361 (77.5)	163 (73.1)	216 (76.1)	110 (79.1)	
**Age**							0.656
<40	621 (44.2)	135 (46.2)	206 (44.2)	103 (46.2)	115 (40.5)	62 (44.6)	
≥40	783 (55.8)	157 (53.8)	260 (55.8)	120 (53.8)	169 (59.5)	77 (55.4)	
**Motive of study**							<0.001
Suspected illness	909 (64.7)	189 (64.7)	257 (55.2)	166 (74.4)	196 (69.0)	101 (72.7)	
Contact tracing	495 (35.3)	103 (35.3)	209 (44.8)	57 (25.6)	88 (31.0)	38 (27.3)	
**History of COVID-19 > 90 days**							0.067
Yes	152 (10.8)	32 (11.0)	64 (13.7)	21 (9.4)	20 (7.0)	15 (10.8)	
No	1252 (89.2)	260 (89.0)	402 (86.3)	202 (90.6)	264 (93.0)	124 (89.2)	
**Professional category**							<0.001
Medical practice	364 (25.9)	101 (34.6)	123 (26.4)	61 (27.4)	56 (19.7)	23 (16.5)	
Nursing	502 (35.8)	84 (28.8)	193 (41.4)	78 (35.0)	96 (33.8)	51 (36.7)	
Auxiliary staff/technicians	298 (21.2)	47 (16.1)	96 (20.6)	40 (17.9)	76 (26.8)	39 (28.1)	
Others ^a^	240 (17.1)	60 (20.5)	54 (11.6)	44 (19.7)	56 (19.7)	26 (18.7)	
**Hospital Areas**							0.040
General Emergency	72 (5.1)	8 (2.7)	28 (6.0)	4 (1.8)	18 (6.3)	14 (10.1)	
Intensive Care Units ^b^	150 (10.7)	32 (11.0)	54 (11.6)	26 (11.7)	29 (10.2)	9 (6.5)	
Hospitalization (Adults and pediatrics)	763 (54.3)	159 (54.5)	253 (54.3)	121 (54.3)	152 (53.5)	78 (56.1)	
Primary health care	152 (10.8)	31 (10.6)	58 (12.4)	26 (11.7)	24 (8.5)	13 (9.4)	
Non COVID-19 hospitalization ^c^	267 (19.0)	62 (21.2)	73 (15.7)	46 (20.6)	61 (21.5)	25 (18.0)	

^a^: Guards, administrative staff, and maintenance staff; ^b^: Intensive Care Units (ICU), Surgical Critical Care Area (SICU) and Operating Rooms; ^c^: Adult Surgical Hospitalization; Administrative Areas; Central Services: Laboratory, Microbiology, Pharmacy, Radiology, Blood bank, etc.; ^d^: The p value of the comparison between health care workers that had completed the vaccination regimen versus those who were unvaccinated.

**Table 2 ijerph-19-13884-t002:** Factors associated with SARS-CoV-2 infection in health care workers (n = 1404).

	COVID-19 Cases (n = 209) n (%)	No COVID-19 Cases (n = 1195) n (%)	Unadjusted OR 95% CI	*p*-Value	Adjusted OR ^a^ 95% CI	*p*-Value
**Vaccinated**						
Full regimen >240 days	53 (25.4)	239 (20.0)	0.66 (0.41–1.07)	0.092	0.63 (0.38–1.05)	0.077
Full regimen 120–240 days	62 (29.7)	404 (33.8)	0.46 (0.29–0.73)	0.001	0.36 (0.22–0.60)	<0.001
Full regimen 12–120 days	6 (2.9)	217 (18.2)	0.08 (0.03–0.20)	<0.001	0.08 (0.03–0.20)	<0.001
Incomplete regimen	53 (25.4)	231 (19.3)	0.68 (0.42–1.11)	0.122	0.65 (0.39–1.08)	0.096
Unvaccinated	35 (16.7)	104 (8.7)	1		1	
**Sex (Female)**	161 (77.0)	903 (75.6)	0.92 (0.65–1.31)	0.648	-	-
**Age (<40)**	85 (40.7)	536 (44.9)	0.84 (0.63–1.14)	0.262	-	-
**Motive of study**						
Suspected illness	84 (40.2)	825 (69.0)	0.30 (0.22–0.41)	<0.001	0.26 (0.19–0.37)	<0.001
Contact tracing	125 (59.8)	370 (31.0)	1		1	
**Previous history of COVID-19 > 90 days**	0 (0.0)	152 (12.7)	Invaluable	-	-	-
**Professional category**						
Medical practice	41 (19.6)	323 (27.0)	0.48 (0.31–0.76)	0.002	0.47 (0.29–0.77)	0.002
Nursing	70 (33.5)	432 (36.2)	0.62 (0.41–0.92)	0.018	0.60 (0.39–0.93)	0.023
Auxiliary staff/technicians	48 (23.0)	250 (20.9)	0.73 (0.47–1.13)	0.159	0.62 (0.39–1.00)	0.005
Others ^b^	50 (23.9)	190 (15.9)	1		1	
**Hospital Areas**						
General Emergency	19 (9.1)	53 (4.4)	2.16 (1.16–4.04)	0.016	2.94 (1.49–5.81)	0.002
Intensive Care Units ^c^	19 (9.1)	131 (11.0)	0.87 (0.48–1.58)	0.655	0.97 (0.52–1.83)	0.925
Hospitalization (Adults and pediatrics)	108 (51.7)	655 (54.8)	0.99 (0.67–1.48)	0.975	1.13 (0.74–1.75)	0.572
Primary health care	25 (12.0)	127 (10.6)	1.19 (0.69–2.06)	0.542	1.34 (0.75–2.41)	0.328
Non COVID-19 hospitalization ^d^	38 (18.2)	229 (19.2)	1		1	

^a^: OR adjusted for vaccination, Motive of study, Professional category and Hospital Areas; ^b^: Guards, administrative staff, and maintenance staff; ^c^: Intensive Care Units (ICU), Surgical Critical Care Area (SICU) and Operating Rooms; ^d^: Adult Surgical Hospitalization; Administrative Areas; Central Services: Laboratory, Microbiology, Pharmacy, Radiology, Blood bank, etc.

**Table 3 ijerph-19-13884-t003:** Adjusted vaccine effectiveness according to the time elapsed since the completion of the vaccination regimen (n = 1404).

	Incomplete	Complete 12–120 Days	Complete 121–240 Days	Complete >241 Days
	aVE ^a^ (IC95%)	aVE ^a^ (IC95%)	aVE ^a^ (IC95%)	aVE ^a^ (IC95%)
**Total**	35.1% (−8.0–60.9%)	91.9% (79.8–96.8%)	63.7% (40.4–77.9%)	37.2% (−5.1–62.5%)
**Sex**				
Male (n = 340)	-	-	-	-
Female (n = 1064)	39.8% (−7.2–66.2%)	89.8% (73.6–96.1%)	66.9% (41.9–81.1%)	45.4% (1.0–69.9%)
**Age**				
<40 (n = 621)	48.6% (−12.2–76.5%)	93.8% (71.4–98.7%)	73.5% (43.5–87.6%)	33.1% (−43.8–68.9%)
≥40 (n = 783)	22.5% (−53.5–60.9%)	90.1% (68.4–96.9%)	56.1% (13.9–77.6%)	39.2% (−23.8–70.1%)
**Motive of study**				
Suspected illness (n = 909)	45.3% (−15.6–74.1%)	92.1% (64.3–98.2%)	31.1% (−38.7–65.8%)	2.0% (−100.6–52.1%)
Contact tracing (n = 495)	36.8% (−37.6–71.0%)	93.4% (77.8–98.0%)	79.8% (57.8–90.4%)	65.0% (23.0–84.1%)

^a^: Vaccine effectiveness adjusted depending on professional category, hospital department and motive of study.

## Data Availability

The data presented in this study are available on request from the corresponding author. The data are not publicly available.

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
