# Peer review of "The Waning of BNT162b2 Vaccine Effectiveness for SARS-CoV-2 Infection Prevention over Time: A Test-Negative Study in Health Care Professionals of a Health Department from January 2021 to December 2021"

_ijerph, 2022, doi:10.3390/ijerph192113884_

Round 1

Reviewer 1 Report

just few gramatical errors and need evidence of pictures

Author Response

Reviewer # 1:

Comments and Suggestions for Authors

Just few gramatical errors and need evidence of pictures.

We appreciate the reviewer's comment regarding the need to correct some grammatical errors, so we have reviewed completely the grammar of the manuscript making the appropriate corrections. Regarding the need for "evidence of pictures", we would like to ask the reviewer for the details of the cases in which he recommends introducing pictures. We remain at your disposal and will be pleased to response any further questions.

Reviewer 2 Report

Jiménez-Sepùlveda and colleagues perform a real-world analyses on vaccine effectiveness of the BNT162b2 vaccine. Despite the novelty of the study is limited by the fact that other similar articles have been published in the last months, it has the merit to collect real-life data from a big cohort of HCP that are, usually, among the most exposed to the virus.

Major comments

Table 2 is in my opinion quite difficult to understand. Is it referred to the number of positive HCP for each subgroup?

On top, is the formatting correct? E.g. the format is very different looking the first 2 columns and comparing them with the 3rd and the 5th)

Similarly, the 4th column seems referred to a p-value

Even the explanation of the table (lines 156-162) is very succinct. I even wonder what “exposure” means in this case, as it is very difficult to measure the exposure to an infection while authors just measure the number of infected individuals (and not, for instance, individuals exposed to the virus but not infected).

Table 3 the test describing results of table 3 is very short and deserves a more comprehensive explanation.

E.g, the age effects is not presented.

Why VE in males has not been calculated?

Why measuring the VE after stratification for “motive of the study”? (I am not saying that is wrong, but authors should explain the rational and present the findings…this is true for every table row)

Similarly, it would be important to show VE also after stratification for other variables (professional category, hospital area and previous COVID history)

However even in this case, an user friendly table easy to be understood also by scientists without a public health backround is desirable.

Minor

Lines 49-50 page 2: the non-effectiveness of non-pharmacological measures is very disputable, I would not enter in this issue

Line 57 page 2: Add the fact that these data are referred to Spain

Lines 82-87 page 2: the verb is missing

Line 92: CRP is PCR?

Line 157 page 4: what is SP?

On the discussion, there are many words with a “-“ in between (e.g, De-Cember line 234, or others )

Author Response

Reviewer # 2:

Comments and Suggestions for Authors

Jiménez-Sepulveda and colleagues perform a real-world analyses on vaccine effectiveness of the BNT162b2 vaccine. Despite the novelty of the study is limited by the fact that other similar articles have been published in the last months, it has the merit to collect real-life data from a big cohort of HCP that are, usually, among the most exposed to the virus.

Major comments:

  1. Table 2 is in my opinion quite difficult to understand. Is it referred to the number of positive HCP for each subgroup? On top, is the formatting correct? E.g. the format is very different looking the first 2 columns and comparing them with the 3rd and the 5th). Similarly, the 4th column seems referred to a p-value

We appreciate the reviewer's comment regarding the problematic interpretation of Table 2. It was indeed a transcription mistake we committed when writing the table, so we have corrected the information included in the first row, improving the interpretability of the table. We have also corrected the legend at the foot of the table. We apologize for the confusion.

  1. Even the explanation of the table (lines 156-162) is very succinct. I even wonder what “exposure” means in this case, as it is very difficult to measure the exposure to an infection while authors just measure the number of infected individuals (and not, for instance, individuals exposed to the virus but not infected).

Although it is true that the term "exposure" could cause confusion, in our manuscript the term "exposure" refers to those healthcare workers who were susceptible to receive a complete vaccination regimen against SARS-CoV-2 infection. In other words, the opportunity to have been properly vaccinated. So, we are not referring to exposure to the virus, but to the vaccine. On the other hand, the way to know if a patient has COVID-19 is through a positive CRP or antigen test.

  1. Table 3 the test describing results of table 3 is very short and deserves a more comprehensive explanation. E.g, the age effects is not presented.

In agreement with the reviewer, we have added a detailed description for table 3.

"Adjusted vaccine effectiveness (aVE) after 12 to 120 days was 91.9% (95% CI: 79.8-96.8), between 121 to 240 days, 63.7% (95% CI: 40.4-77.9) and after 241 days, 37.2% (95% CI: -5.1-62.5). The majority of participants were women (1064 (75.8%)), showing a decrease in aVE after 12 to 120 days since the second dose administration from 89.8% (73.6% - 96.1%) to 45.4% (1.0% - 69.9%) after more than 241 days had elapsed. Likewise, those older than 40 years (783 (55.8%)) showed a decrease in aVE from 90.1% (68.4% - 96.9%) at 12 to 120 days after administration of the complete regimen to 39.2% (-23.8% - 70.1%) after 241 days. Regarding the motive of study, 909 (64.7%) HCWs were under suspected disease and showed a decrease of aVE from 92.1% (64.3% - 98.2%) to 2.0% (-100.6% - 52.1%) for the periods between 12 to 120 days and after 240 days, respectively".

  1. Why VE in males has not been calculated?

 In case-control studies, variables that have only two categories and one of the two is considered complementary, only the reference variable (female/male) is expressed. In the case of variables that are not complementary, even if they have two categories, both are indicated (suspicion of disease/contact study).

  1. Why measuring the VE after stratification for “motive of the study”? (I am not saying that is wrong, but authors should explain the rational and present the findings…this is true for every table row).

 Our objective in Table 3 was to evaluate whether stratifying aVE by groups, statistically significant differences existed in terms of sex, age, and motive of the study in HCWs. Those included due to close contact with a confirmed case had similar characteristics to those of an asymptomatic patient, and to those in whom symptoms were suspected. Therefore, our objective was to determine whether there was any pattern for aVE behavior in these specific groups. Likewise, in accordance with the reviewer's recommendations, we have included in the results the evolution over time of aVE depending on the motive for study.

  1. Similarly, it would be important to show VE also after stratification for other variables (professional category, hospital area and previous COVID history). However even in this case, an user friendly table easy to be understood also by scientists without a public health background is desirable.

 Indeed, we have not included other variables for analysis because we do not have sufficient data to allow us to perform the statistical analysis. On the other hand, unlike sex, age and the motive for study; professional category, the department, etc, do not biologically predispose HCWs to develop SARS-CoV-2 infection. However, not including these variables could be a limitation of the study. Therefore, we added this sentence to the discussion:

"Finally, we did not include in the analysis of aVE (table 3) the variables: professional category, hospital department and previous COVID-19 infection because we did not have enough data for their statistical analysis, which could be a limitation. However, these variables do not biologically predispose to SARS-CoV-2 infection in healthcare workers ".

Minor comments:

  1. Lines 49-50 page 2: the non-effectiveness of non-pharmacological measures is very disputable, I would not enter in this issue.

In agreement with the reviewer, we have replaced the text "were not effective" with "were not enough". (Line 104)

  1. Line 57 page 2: Add the fact that these data are referred to Spain.

According to the reviewer We include in the introduction (line 112) that the data referred to in the text correspond to Spain.

  1. Lines 82-87 page 2: the verb is missing.

 According to the reviewer observations, we corrected grammatical errors in the manuscript.

  1. Line 92: CRP is PCR?

We have replaced PCR for CRP, meaning C-reactive protein.

  1. Line 157 page 4: what is SP?

We committed a transcription mistake, SP stands for HCWs and we have corrected this in the manuscript.

  1. On the discussion, there are many words with a “-“ in between (e.g, De-Cember line 234, or others ).

We appreciate the reviewer's comment regarding the revision of some grammatical errors, so we will proceed to review the grammar of the manuscript again and make the pertinent corrections.

Author Response

Reviewer # 3: PDF comments

Comments and Suggestions for Authors

  1. The manuscript needs to be proofread and checked for grammar. Example: Lines 63-65 on page 2 read “These types of studies are essential to generate evidence for continuous evaluation of vaccines benefits and risks and to improve decision-making about their use”. It should read: “These types of studies are essential to generate evidence for the continuous of vaccine benefits and risks and improved”.

 According to the reviewer observations, we corrected grammatical errors in the manuscript.

  1. Table 2 is quite confusing. Columns labeled “incomplete” and “Unvaccinated” purport in the column headings to give the number and percentage of infected, but the actual numbers in the columns appear to only be the percentages.

We appreciate the reviewer's comment regarding the difficult interpretation of Table 2. It was indeed a transcription mistake in the table, so we have included the correct data in the first row, which means that the table can now be interpreted correctly. We also corrected the legend corresponding to the foot of the table. We apologize for the confusion.

Line 127-128 “the unadjusted odds ratio (OR) an adjusted odds ratio (ORa) were calculated using a logistic regression model”. What were the variables used in the logistic regression model?.

Indeed, we used a logistic regression model. The variables included in the analysis were as indicated at the bottom of the table 2: Motive of study, Professional category and Hospital Areas.

  1. A Benford’s Law Chi-Square Test of the first digits of the non-zero reported figures reveals a substantial departure of the reported statistics at the from Benford’s Law at the 0.7081% level. I am not accusing the authors of fraud, but I would like to see their data,please. I would also recommend the authors check their statistics.

In agreement with the reviewer, we would like to know the exact data you would like to review and how you would like us to get it to you. We remain at your disposal.

I would like to know what logistic model they use to figure the odds ratio.

We used a logistic regression model where, firstly, a bivariate model was used to calculate the crude OR for each of the variables and, secondly, for adjusted OR analysis, those variables that had shown statistically significant differences in bivariant analysis were introduced.

Round 2

Reviewer 2 Report

If I understood correctly, authors defined a case based on the level of C-reactive protein (which is not specific at all for COVID but is a general sign of inflammation). This is a great limitation that might invalidate the whole study. Authors should carefully check this and provide details of the methods. Similarly, it’s not clear what they mean for “antigenic test”. Are these rapid tests ? detecting what? Form which specimens?.

From the other side, authors have addressed several concerns.

However, several doubts still persist:

Table 2

The percentage close to absolute numbers -e.g. COVID cases: 53 (25.4) - is calculated on the total of “covid cases” or on “NO covid cases”.

In my opinion this is not informative (the percentage is influenced by the total number of each subgroup).

It would be much more informative to calculate the % inside subgroups.

Es, among the “full regimes >240 days”: the 53 covid cases represent the 18% while he no covid cases represent the 82%.

Authors say:

 In case-control studies, variables that have only two categories and one of the two is considered complementary, only the reference variable (female/male) is expressed. In the case of variables that are not complementary, even if they have two categories, both are indicated (suspicion of disease/contact study).

But why the calculated VE for both >40 and <40?

Authors say:

On the other hand, unlike sex, age and the motive for study; professional category, the department, etc, do not biologically predispose HCWs to develop SARS-CoV-2 infection. However, not including these variables could be a limitation of the study.

This is not true at all: there are professional categories that are more exposed to the virus for the type of patients they visit. Please rephrase the sentence

Author Response

Reviewer # 2:

Comments and Suggestions for Authors

  1. If I understood correctly, authors defined a case based on the level of C-reactive protein (which is not specific at all for COVID but is a general sign of inflammation). This is a great limitation that might invalidate the whole study. Authors should carefully check this and provide details of the methods.

 We appreciate the reviewer's comment. Indeed, during the first revision a change in the spelling of the abbreviation "CPR" was suggested. When correcting this mistake by the automated search and correction, we committed the error of writing "CRP" in the entire text. We clarify that in our study we used the polymerase chain reaction test for the diagnosis of active infection and not the C-reactive protein, so we amend this mistake by changing all the misspelled abbreviations for the correct one: PCR. We apologize for the misunderstanding and thank you for the observation. 

Similarly, it’s not clear what they mean for “antigenic test”. Are these rapid tests? detecting what? Form wich specimens?

We refer to antigen test in the manuscript as one of the viral tests used for the diagnosis of COVID-19. It is an immunoassay test that detects the presence of a specific viral antigen, indicating acute infection and having similar specificity but lower sensitivity than most nucleic acid amplification tests (polymerase chain reaction, reverse transcription, etc)1. Antigen tests produce results rapidly (in approximately 15-30 minutes) and are commonly used in the diagnosis of other respiratory pathogens, including influenza viruses and respiratory syncytial virus (RSV)2

  1. Overview of Testing for SARS-CoV-2, the virus that causes COVID-19. Centers for Disease Control and Prevention. https://www.cdc.gov/coronavirus/2019-ncov/hcp/testing-overview.html
  2. Guidance for antigen testing for SARS-CoV-2 for healthcare providers testing individuals in the community. Centers for Disease Control and Prevention. https://www.cdc.gov/coronavirus/2019-ncov/lab/resources/antigen-tests guidelines.html

  1. Table 2. The percentage close to absolute numbers -e.g. COVID cases: 53 (25.4) - is calculated on the total of “covid cases” or on “NO covid cases”. In my opinion this is not informative (the percentage is influenced by the total number of each subgroup). It would be much more informative to calculate the % inside subgroups. Es, among the “full regimes >240 days”: the 53 covid cases represent the 18% while he no covid cases represent the 82%.

As a matter of fact, in Table 2 there are two columns, the first including cases named: "COVID-19 cases" and the second including controls, named: "No COVID-19 cases". In case-control studies with negative test, such as ours, the proportions are reported by columns, so that we can know how many subjects are included within each group (cases and controls), depending on their vaccination dose regimen. 

In regard to the example given by the reviewer: 53 of the 209 cases, correspond to 25.4%. This percentage reflects the total number of HCWs exposed to the complete vaccination regimen, more than 240 days after the administration of the complete regimen. Finally, we are not representing the proportions in rows because that is typical of cohort studies.

  1. Authors say: “ In case-control studies, variables that have only two categories and one of the two is considered complementary, only the reference variable (female/male) is expressed. In the case of variables that are not complementary, even if they have two categories, both are indicated (suspicion of disease/contact study)”. But why the calculated VE for both >40 and <40?

We appreciate the reviewer's comment. As specified in the reply to the first review, the reference variable is not indicated, however, it is indeed calculated and also used to obtain the vaccine effectiveness (VE). In the case of age, both ranges are specified because the variables that have intervals might be "less or equal than" and therefore it cannot be deduced whether the other range includes the value in between the two intervals.

  1. Authors say: “On the other hand, unlike sex, age and the motive for study; professional category, the department, etc, do not biologically predispose HCWs to develop SARS-CoV-2 infection. However, not including these variables could be a limitation of the study”. This is not true at all: there are professional categories that are more exposed to the virus for the type of patients they visit. Please rephrase the sentence.

We appreciate the reviewer's comment and apologize for the confusion that our previous reply might have generated. Indeed, the professional category influences on SARS-CoV-2 virus exposure (for example, in the case of nurses who have had closer contact with patients), however, in our study when we refer to "exposure" we mean the fact of being vaccinated (or exposed to the vaccine) against SARS-CoV-2 infection. For this reason, we do not consider the professional category as a biological determinant for VE, since the immune response against SARS-CoV-2 infection of each subject is not influenced by the job performed or the area in which he/she works.

Reviewer 3 Report

It looks good.

Author Response

Thank you very much for your support